# Microalgae and Bacteria Interaction—Evidence for Division of Diligence in the Alga Microbiota

Yekaterina Astafyeva,[a] Marno Gurschke,[a] Minyue Qi,[b] Lutgardis Bergmann,[a] Daniela Indenbirken,[c] Imke de Grahl,[d] Elena Katzowitsch,[e] Sigrun Reumann,[d] Dieter Hanelt,[f] Malik Alawi,[b] ⓘ Wolfgang R. Streit,[a] ⓘ Ines Krohn[a]

[a]University of Hamburg, Institute of Plant Science and Microbiology, Department of Microbiology and Biotechnology, Hamburg, Germany
[b]University Medical Center Hamburg-Eppendorf, Bioinformatics Core, Hamburg, Germany
[c]Heinrich-Pette-Institute, Leibniz Institute for Experimental Virology, Virus Genomics, Hamburg, Germany
[d]University of Hamburg, Institute of Plant Science and Microbiology, Department of Plant Biochemistry and Infection Biology, Hamburg, Germany
[e]University of Würzburg, Core Unit Systems Medicine, Würzburg, Germany
[f]University of Hamburg, Institute of Plant Science and Microbiology, Department of Aquatic Ecophysiology and Phycology, Hamburg, Germany

**ABSTRACT** Microalgae are one of the most dominant forms of life on earth that is tightly associated with a distinct and specialized microbiota. We have previously shown that the microbiota of *Scenedesmus quadricauda* harbors less than 10 distinct microbial species. Here, we provide evidence that dominant species are affiliated with the genera of *Variovorax*, *Porphyrobacter*, and *Dyadobacter*. Experimental and transcriptome-based evidence implies that within this multispecies interaction, *Dyadobacter* is a key to alga growth and fitness and is highly adapted to live in the phycosphere. While presumably under light conditions the alga provides the energy source to the bacteria, *Dyadobacter* produces and releases mainly a large variety of polysaccharides modifying enzymes. This is coherent with high-level expression of the T9SS in alga cocultures. The transcriptome data further imply that quorum-quenching proteins (QQ) and biosynthesis of vitamins $B_1$, $B_2$, $B_5$, $B_6$, and $B_9$ are expressed by *Dyadobacter* at high levels in comparison to *Variovorax* and *Porphyrobacter*. Notably, *Dyadobacter* produces a significant number of leucine-rich repeat (LRR) proteins and enzymes involved in bacterial reactive oxygen species (ROS) tolerance. Complementary to this, *Variovorax* expresses the genes of the biosynthesis of vitamins $B_2$, $B_5$, $B_6$, $B_7$, $B_9$, and $B_{12}$, and *Porphyrobacter* is specialized in the production of vitamins $B_2$ and $B_6$. Thus, the shared currency between partners are vitamins, microalgae growth-promoting substances, and dissolved carbon. This work significantly enlarges our knowledge on alga-bacteria interaction and demonstrates physiological investigations of microalgae and associated bacteria, using microscopy observations, photosynthetic activity measurements, and flow cytometry.

**IMPORTANCE** The current study gives a detailed insight into mutualistic collaboration of microalgae and bacteria, including the involvement of competitive interplay between bacteria. We provide experimental evidence that Gram-negative bacteria belonging to the *Dyadobacter*, *Porphyrobacter*, and *Variovorax* are the key players in a *Scenedesmus quadricauda* alga-bacteria interaction. We impart strong evidence that *Dyadobacter* produces and releases polysaccharides degradation enzymes and leucine-rich repeat proteins; *Variovorax* supplies the consortium with auxins and vitamin $B_{12}$, while *Porphyrobacter* produces a broad spectrum of B vitamins. We show not only that the microalgae collaborate with the bacteria and vice versa but also that the bacteria interact with each other via quorum-sensing and secretion system mechanisms. The shared currency between partners appears to be vitamins, microalgae growth-promoting substances, and dissolved carbon.

**KEYWORDS** microalgae and microbiota interaction, synthetic plant-bacteria system, phycosphere biofilm, *Scenedesmus quadricauda*, *Dyadobacter* sp.

Address correspondence to Ines Krohn, ines.krohn@uni-hamburg.de.

The authors declare no conflict of interest.

The term microalgae is a collective phylogenetical expression for a highly heterogeneous group of eukaryotic and prokaryotic microorganisms that are unicellular photosynthetic organisms growing in marine and aquatic environments, respectively (1, 2). In general, microalgal cells are surrounded by a phycosphere, which is a region rich in organic material released by the algae creating the key interface in which algae and other organisms tightly interact (3). Thus, algae and bacteria synergistically affecting each other's physiology and metabolism together have an impact on the ecosystems and represent all modes of interactions between various organisms, ranging from mutualism to parasitism (4, 5).

Tight associations of microalgae and bacteria have resulted in the evolution of a complex network of these cross-kingdom interactions and a potential specialization of various organisms. Orchestrated nutrient exchange, mutual support of growth factors, and quorum-sensing mediation define a wide spectrum of associations at highly complex assemblages of unicellular microalgae and associated bacteria (3). The positive effects of algae-bacteria interaction on algal growth and flocculation processes has changed the scenario of considering bacteria as mere contamination of algae cultures (6). Several studies have verified that the microbiome of microalgae consists mainly of species of the phyla Bacteroidota, Flavobacteria, Alphaproteobacteriota, Betaproteobacteriota, and Gammaproteobacteriota (2, 7). Many noncultivatable organisms can be detected as well, but the overall abundance of microorganisms is often limited to approximately 30 species (2, 8). It is still unclear whether all of these bacteria are important for algal health and survival of the community. Interestingly, limited potential invasive or pathogenic species have been also detected during the screening but somehow appear to live in a balanced manner in the community, which can be explained by a selection pressure that favors the microalgae-bacteria consortia (9–12).

However, for applications in aquaculture, the potential of the interactions between microalgae and microorganisms may improve algal biomass production and may enrich the biomass with compounds of biotechnological interest such as lipids, carbohydrates, and pigments. Consequently, growth and photosynthetic activity of bacteria profit from the interaction with microalgae, especially by adhesion, clumping factor, motility, chemotaxis, secretion systems, quorum-sensing (QS), and quorum-quenching (QQ) systems and synthesis of growth promoters (5, 13–15). Intra- and interspecies communication among microbes occurs via a complex set of signal molecules secreted during both beneficial and harmful interactions that coordinate and control the behavior of microorganisms in mixed communities. Production and release of signal molecules will be recognized by other members of the microbial community, will initiate the communication causing up- or downregulation of gene expression, and will alter the activity and physiology of the recipient (8, 16, 17). Bacterial communication provides the regulation of virulence-associated factors, propagation, population density, and change in metabolic rate, various bacterial functions, including biofilm formation, motility, adhesion, and coordination between microbial communities (18).

The goal of the study was detailed insight into mutualistic collaboration of microalgae and bacteria, including the involvement of competitive interplay between bacteria. Since not much is known about the individual role of the individual members of the multispecies consortia, we set out to determine the role of one of the bacterial isolates from nonaxenic algal culture, *Dyadobacter* sp. HH091, in the interplay with the other dominant bacterial species and the alga. For this purpose, we isolated the bacterium from the microbiome, made it genetically accessible, and studied its impact using physiological and omics-based methods.

## RESULTS

Previous studies have shown that microalgae obtained from strain collections are associated with various bacterial communities with a rather low diversity (2, 8). In this research, we concentrated on the microalga *Scenedesmus quadricauda* and its bacterial community. In this community, we observed bacteria affiliated with the genera *Dyadobacter* (phylum

Bacteroidota) and *Variovorax* and *Porphyrobacter* (phylum Proteobacteriota) as the dominant species (2, 9, 19). To gain insight into the physiological role of the microbiota, we set out to isolate individual strains and used them in cocultivation studies. Using classical enrichment strategies, we were able to cultivate a single *Dyadobacter* affiliated with a *S. quadricauda* laboratory nonaxenic culture on tryptone yeast extract salts (TYES) medium as described in the Materials and Methods section. The isolation of *Porphyrobacter* and *Variovorax* from the same *S. quadricauda* laboratory nonaxenic culture was unsuccessful.

The obtained *Dyadobacter* isolate was designated *Dyadobacter* sp. HH091 (from here on called "HH091"). Its phylogenetic affiliation with the genus *Dyadobacter* (phylum Bacteriodota) was initially verified by using 16S rRNA gene amplification and DNA sequencing. Following this, the organism's chromosomal DNA was extracted and sent out to establish its genome sequence. The HH091 genome draft consisted of 30 contigs and has a size of 7,837,776 bp with a G+C content of 43.87% (Table S1; Fig. S1). Annotation identified 6,628 genes, 6,565 of which are protein coding (Table S1). The HH091 genome was deposited at Integrated Microbial Genomes & Microbiomes (IMG/M) (https://img.jgi.doe.gov) under accession IMG ID 2842103827.

*Dyadobacter* is well adapted for life in multispecies communities and in the vicinity of microalga, which is confirmed by the metatranscriptomic activity of genes affiliated with competitive and plant-bacteria interaction. Its genome codes for several fascinating genetic features, such as multiple QS and QQ loci, which play important roles in cell signaling, and surviving in competitive conditions. Notably, we observed 23 possible QQ genes and 41 *luxR* solos. Further, it codes for complete secretion systems of type 4 (T4SS), 5 (T5SS), 6 (T6SS), and 9 (T9SS). In addition, a remarkable wealth of glycosyl hydrolases (GHs) was observed. A total of 69 GHs, 47 polysaccharide lyases (PLs) and 50 carbohydrate esterases (CEs) were predicted. Table 1 and Tables S1 to S5 give an overview of the genomic features of HH091.

**Dynamics of the bacterial colonization of the microalgae studied by confocal microscopy.** Based on the above-made observations, we were interested to analyze the effects of HH091 on the microalga in cocultures (Fig. 1). We were able to electroporate and stably maintain the plasmid pBBR1MCS-5-eGFP in HH091. This plasmid carries an enhanced green fluorescent protein (eGFP) under the control of the lac promoter and allowed the detection of HH091 on the surface and inside the algal cells using confocal laser scanning microscope (CLSM). In our experiments, we could not observe any growth inhibition of *Dyadobacter* sp. HH091 (wild type [WT]) and the plasmid-expressing bacteria. Additional high-resolution CLSM images of *S. quadricauda* incubated with HH091 implied that the bacterium was often tightly associated with the algal cells (Fig. 1). Fig. 1C presents the middle layer of a Z-Steck image of *S. quadricauda* with HH091, in which the bacteria is found inside the algal cells. We examined cocultures of HH091 grown together with *S. quadricauda* and compared its photosynthetic activity and relative fitness with the antibiotic-treated algal control cultures over a time period of 13 days (Fig. 2A). To identify the difference in the growth of algal cultures (with and without HH091), we used the optical density measurement (Fig. 2A). In these tests, the first hints of visible difference were observed after 2 to 3 days. The photosynthetic activity measurements by pulse-amplitude-modulation (PAM) fluorometry demonstrated that the coculturing of *S. quadricauda* with HH091 resulted in 1.5-fold increase of the optimal quantum yield of Photosystem II (PS II) in comparison to control antibiotic-treated microalgae (Fig. 2A). Additional microscopic and fluorescence-activated cell sorting (FACS) analyses verified that the number of viable and photosynthetic active cells in the presence of HH091 was much higher in comparison to antibiotic-treated cultures (Fig. 2B and C). In these tests, the algae culture (with and without HH091) were analyzed for 13 days after inoculation and by monitoring the chlorophyll autofluorescence of the algal population or the low-level autofluorescence of the bacteria in the FL2 channel (Fig. 2B). In a FSC X FL2 density blot diagram, three specific populations were detectable. These populations could be assigned as follows: population I, bacteria; population II, dead and lysed algal cells; and; population III, healthy microalgae (Fig. 2B; Fig. S2). In the beginning of the experiment, the content of living algal

**Table 1** Possible interaction pathways of *Dyadobacter* sp. HH091 genome[a]

| Selected key features | *Dyadobacter* sp. HH091 |
|---|---|
| Transporter, efflux pumps, and secretion systems | |
| Transport proteins and efflux pumps | 138 |
| MFS, ABC, and biopolymer transporters | 189 |
| T4SS | 3 |
| T5SS | 9 |
| T6SS | 9 |
| T9SS | 41 |
| Sec-independent protein secretion pathway | 2 |
| Sec-SRP | 5 |
| | |
| Signal transduction and regulation mechanisms | |
| Response regulators (NarL/FixJ, LytR, OmpR) | 116 |
| ECF sigma factors | 54 |
| | |
| Polysaccharides degradation | |
| Auxiliary activities | 16 |
| Carbohydrate esterases | 50 |
| Glycoside hydrolases | 69 |
| Glycosyl transferases | 69 |
| Polysaccharide lyases | 47 |
| Carbohydrate-binding modules | 21 |
| Peptidases | 139 |
| | |
| Competitive interactions | |
| Potential antibiotic substances | 16 |
| Endonucleases and exonucleases | 60 |
| Permeases | 157 |
| Proteases | 69 |
| Heme synthesis | 9 |
| Quorum quenching | 23 |
| | |
| Bacteria-plant interaction pathways | |
| Vitamins biosynthesis | 224 |
| Invasion-associated proteins | 2 |
| LRR proteins | 2 |
| ROS tolerance | 13 |

[a]The table shows the key features of possible competitive and plant-bacteria interaction pathways of the *Dyadobacter* sp. HH091 genome, using Integrated Microbial Genomes (IMG) function search. The data are shown as the total number of hits. Major facilitator superfamily (MFS), ATP-binding cassette (ABC)-transporters, type 4 secretion system (T4SS), type 5 secretion system (T5SS), type 6 secretion system (T6SS), type 9 secretion system (T9SS), secretion (Sec), secretion-signal recognition particle (Sec-SRP), extracytoplasmic function (ECF), leucine-rich repeat (LRR), reactive oxygen species (ROS).

cells in an antibiotic-treated culture was 59.4% ($\pm$0.5), whereas 5.56% ($\pm$0.6) of the population included bacteria with 13.2% ($\pm$0.8) of lysed algal cells. In the end of the experiment, we identified 70.8% ($\pm$1.6) of healthy microalgae, 18.8% ($\pm$0.5) of lysed algal cells, and 3.49% ($\pm$0.5) of bacteria in a coculture with *Dyadobacter*. In general, we

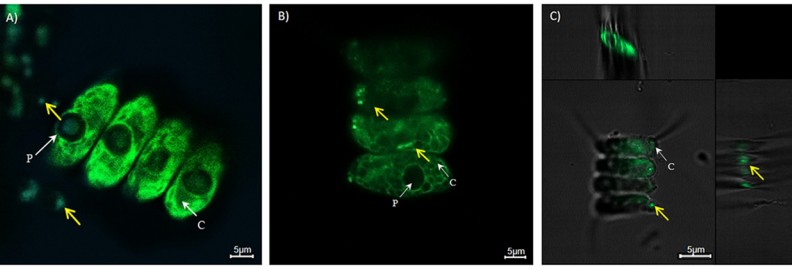

**FIG 1** Confocal microscope images of the strain *Dyadobacter* sp. HH091 expressing enhanced green fluorescent protein (eGFP) (yellow arrows) in coculture with *S. quadricauda* MZCH 10104. A confocal laser scanning microscope (CLSM) Axio Observer.Z1/7 LSM 800 (Carl Zeiss Microscopy GmbH, Jena, Germany) with ZEN software (version 2.3; Carl Zeiss Microscopy GmbH) was used. (A) Three-day culture. (B) seven-day culture; (C) seven-day culture, Z-Steck image. An autofluorescence quenching kit was used to lower the autofluorescence of chlorophyll of the microalga. c = chloroplast; p = pyrenoid. Bar = 5 $\mu$m.

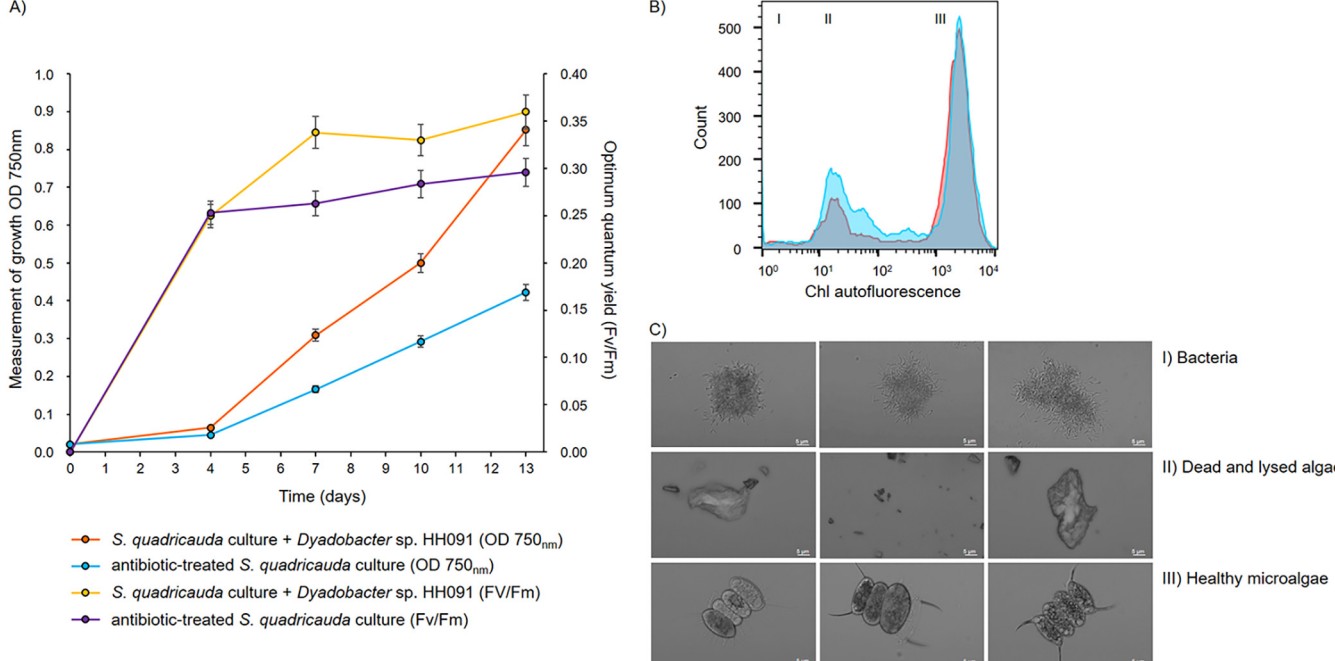

**FIG 2** (A, B) Photosynthetic activity and growth measurement (A) and fluorescence-activated cell sorting (FACS) analyses (B) of *S. quadricauda* MZCH 10104 in coculture with the strain *Dyadobacter* sp. HH091. (A) Increased photosynthetic activity and growth rate (optical density at 750 nm [OD 750 nm]) can be observed in the coculture with HH091. (B) Comparison of populations, based on chlorophyll intensity. Orange shading indicates a stable distribution of bacteria (population I), dead and lysed algae (population II), and healthy microalgae (population III) in the coculture at the end of the experiment (13 days). Blue shading shows how the proportion of dead and lysed algae is increased in the culture without HH091. The raw data are available in Fig. S2. (C) Images of cell sorting acquired with CLSM Axio Observer.Z1/7 LSM 800 (Carl Zeiss Microscopy GmbH, Jena, Germany) and ZEN software (version 2.3; Carl Zeiss Microscopy GmbH). Bar = 5 $\mu$m. Pulse-amplitude-modulation (PAM) fluorometry and FACS analyses demonstrated the improved fitness of *S. quadricauda* cocultured with HH091.

count approximately 10 to 15 bacterial cells for 1 microalgal cell. Comparison of populations, based on chlorophyll intensity, indicates a stable distribution of bacteria (population I), dead and lysed algae (population II), and healthy microalgae (population III) in the coculture at the end of the experiment (13 days), whereas in the culture without HH091, the proportion of dead and lysed algae is increased (3.24% of bacteria [population I], 27.4% of lysed algal cells [population II], and 57.7% of algae cells [population III]) (Fig. 2B; Fig. S2).

**Transcriptome sequencing (RNA-Seq) global analysis of multispecies bacterial consortia and microalga transcriptomes.** We hypothesize that more than one compound is essential for a healthy algae growth, which is not known yet. To further analyze the role of HH091 in this specific cross-kingdom interaction, we set out to analyze the transcriptomes of this bacterium in the background of the native multispecies microbiota. We chose to use the original *S. quadricauda* lab culture, from which HH091 was isolated, as this would most likely resemble the native situation in a multispecies community. Thus, we analyzed the multispecies transcriptome of the microbiota at exponential and stationary growth phases of *S. quadricauda* cultures (Table S6; Fig. 3). Sequences obtained for this study were submitted to the European Nucleotide Archive (ENA; PRJEB23338).

In total, we obtained 42 million (mio) reads of bacteria data after trimming. The data are the results of three replicates with each replicate producing between 19 and 25 mio reads. The trimmed reads were assembled into contigs for the exponential and stationary phase experiments (Table S6). In total, the RNA-Seq data covered a significant portion of overall bacterial genomes and the affiliated pathways.

Fig. 3 summarizes the transcriptomics for the exponential (A) and the stationary (B) growth phases. In all treatments, the largest fraction of transcribed genes had no function assigned (20 to 25%). In the bacteria, the second largest fraction was associated

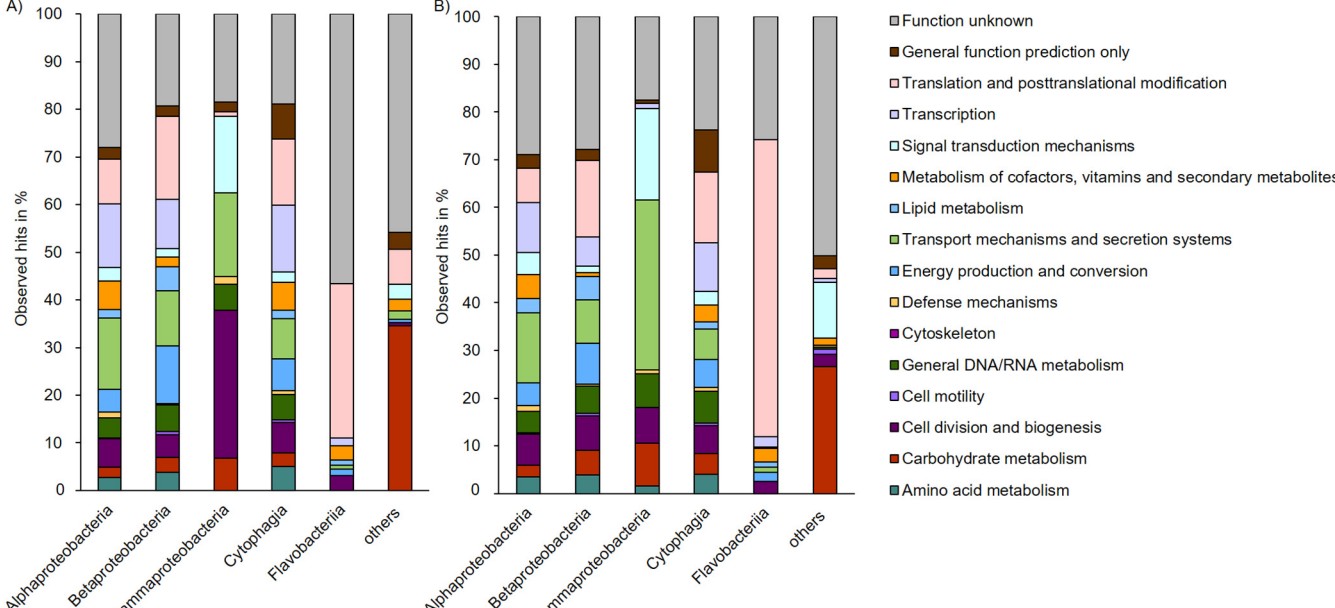

**FIG 3** Expressed genes of bacteria in coculture with the microalga *S. quadricauda* during the exponential (A) and stationary (B) growth phases. The generated sequence data (approximal 42 mio reads) were mapped to the available genomes and metagenomes (8). The most dominant species of the microbiome are affiliated with the phyla of the Proteobacteriota (*Variovorax* and *Porphyrobacter*) and Bacteroidota (*Dyadobacter*).

with transport mechanisms and secretion systems during the exponential phase (15% Alphaproteobacteriota, 11.5% Betaproteobacteriota, 17.5% Gammaproteobacteriota, 8.5% Cytophagia, 0.73% Flavobacteriia, and 1.64% others). Further, a large fraction was connected to translation and posttranslational modification (9.5% Alphaproteobacteriota, 17% Betaproteobacteriota, 1% Gammaproteobacteriota, 14% Cytophagia, 32.5% Flavobacteriia, and 7.5% others). Notably, the bacteria showed relatively high levels of transcription of genes linked to the biosynthesis of cofactors, vitamins, and secondary metabolites (6% Alphaproteobacteriota, 2% Betaproteobacteriota, 6% Cytophagia, 3% Flavobacteriia, and 2.5% others).

The transcriptionally most active bacteria were *Dyadobacter* (3 mio reads mapped), *Porphyrobacter* (2.4 mio reads mapped), and *Variovorax* (1 mio reads mapped) at the exponential growth phase. This was similar at the stationary growth phase with *Dyadobacter* (6.9 mio reads), *Porphyrobacter* (5.5 mio reads), and *Variovorax* (1.3 mio reads). The data sets were normalized for the further analyses as outlined in the Materials and Methods section.

**RNA-Seq identifies highest bacterial transcribed genes related to carbohydrate degradation, competitive, and plant-bacteria interaction.** In the following, we highlight some of the most relevant findings of the whole data sets. Notably, we included those genes that had more than 500 counts/gene. The detailed analyses of the most strongly expressed genes are summarized in Fig. 4. The most strongly expressed genes affiliated with *Dyadobacter* were related to carbohydrate degradation, competitive, and plant-bacteria interaction. Genes, most strongly expressed in *Porphyrobacter* and *Variovorax*, included competitive interaction mechanisms, vitamins biosynthesis, secretion systems, and fatty acids biosynthesis (Fig. 4).

Nevertheless, the transcriptome data set hinted toward a metabolic symbiosis between microalgae/bacteria and bacteria/bacteria. Fig. 4 reflects the expression of genes affiliated with transporter and secretion systems, signal transduction and regulation mechanisms, polysaccharide degradation, competitive interactions, and interaction pathways of the microbiome of *S. quadricauda* during the exponential (A) and the stationary (B) growth phases.

**Correlation between transcriptome and genome analyses exposes multifaceted synergistic cooperation.** To examine the distribution of specific protein families across the transcriptionally most active bacterial genomes, we performed a protein

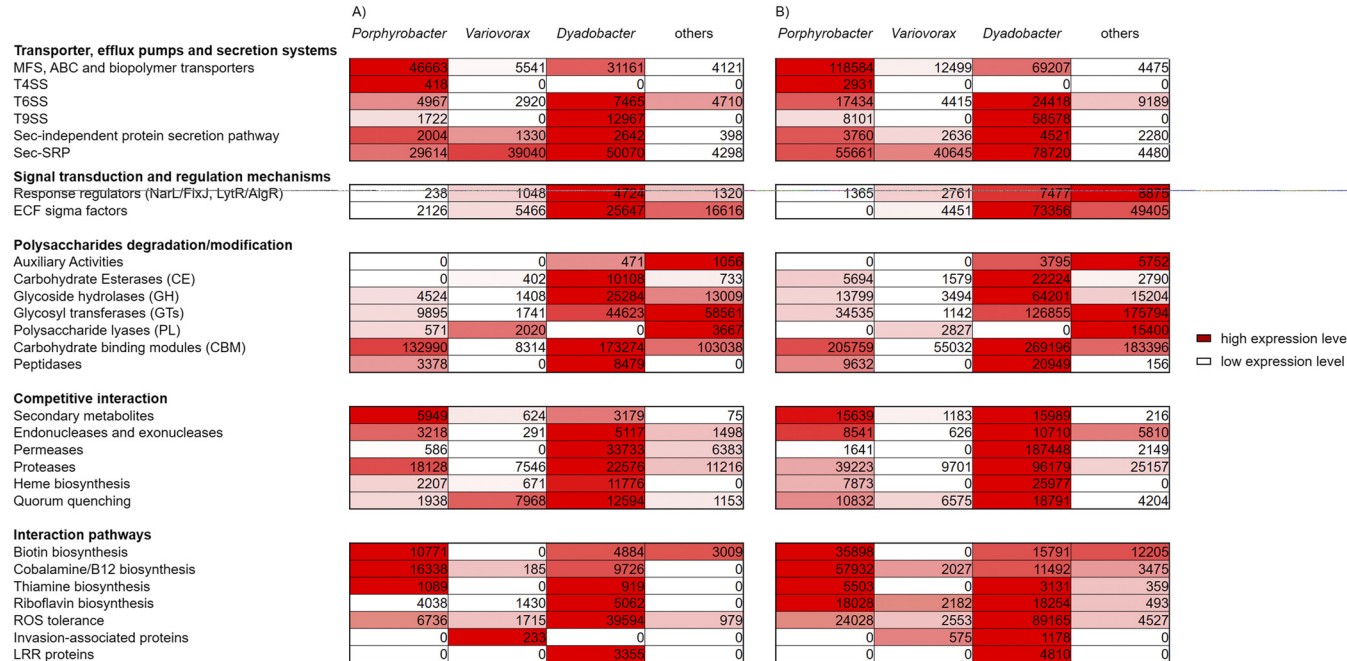

**FIG 4** Heat map reflecting the relative expression of genes affiliated with overall plant-bacteria interaction pathways of the bacterial metagenome of *S. quadricauda* during the exponential (A) and stationary (B) growth phases. Red shading indicates a high expression level, and white indicates a low expression level. The numbers indicate the total amount of hits during RNA-Seq. Genes have more than 500 counts/gene.

family comparison across the microbiome of *S. quadricauda* MZCH 10104 (Table S7). The protein families were filtered in affiliation with overall plant-bacteria interaction pathways.

Crucial key features of overall plant-bacteria interaction pathways highlighted in metatranscriptomic analysis were mapped to the reference genomes of *Porphyrobacter* sp. AAP82 (IMG 2551306481), *Variovorax paradoxus* S110 (IMG 644736413), and *Dyadobacter* sp. HH091 (IMG 222279) (Fig. 5). For *Porphyrobacter* and *Variovorax*, we used publicly available reference genome IMG 2551306481 (GenBank accession number ANFX00000000) and IMG 644736413 (GenBank accession numbers NC012791 and NC012792), respectively, and mapped the transcriptome data on these genomes. These two genomes were chosen as reference genomes because they have high-quality annotations and are originally isolated from aquatic/plant habitats (20, 21). The transcriptome data obtained for *Dyadobacter* were mapped onto the genome of HH091 (IMG 222279). The circular mapping was generated using the Circular Genome Viewer tool within PATRIC, the Pathosystems Resource Integration Center (www.patricbrc.org). Moving inward, the subsequent two rings show coding DNA sequences (CDSs) in forward (brown) and reverse (green) strands. Blue and yellow plots indicate GC content and a GC skew [(GC)/(G+C)]. Key features of possible competitive and plant-bacteria interaction pathways are marked with the following colors: purple for transporter, efflux pumps, and secretion systems; orange for signal transduction and regulation mechanisms; turquoise for polysaccharide degradation; red for competitive interactions; and blue for bacteria-plant interaction pathways (Fig. 5).

**B vitamins are key drivers in bacteria-alga and bacteria-bacteria interactions.** Thiamin biosynthesis genes were mainly expressed in *Dyadobacter* involving the whole cluster of required biosynthesis genes. *Dyadobacter* and *Porphyrobacter* are most likely auxotrophic for $B_{12}$ biosynthesis (2). Biosynthesis of cobalamin requires approximately 30 enzymatic steps for its complete *de novo* construction. Among two existing distinct biosynthetic pathways, which are termed the aerobic and anaerobic routes (22), *Variovorax* codes in its genome for the anaerobic pathway of the vitamin $B_{12}$ biosynthesis. Thus, it is likely that vitamin $B_{12}$ was provided by *Variovorax* and not by any of the two other bacteria, as they lack these genes. A riboflavin ($B_2$) biosynthesis cluster is

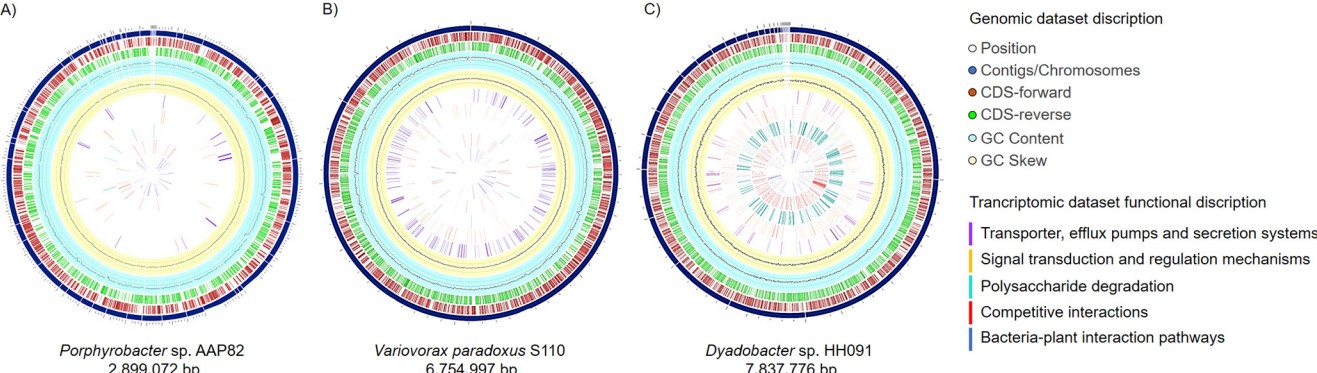

**FIG 5** Metatranscriptome mapping to the reference genomes of *Porphyrobacter* sp. AAP82 (IMG 2551306481) (A), *Variovorax paradoxus* S110 (IMG 644736413) (B), and *Dyadobacter* sp. HH091 (IMG 222279) (C). Moving inward, the subsequent two rings show coding DNA sequences (CDSs) in forward (brown) and reverse (green) strands. Blue and yellow plots indicate GC content and a GC skew ([GC]/[G+C]). Key features of possible competitive and plant-bacteria interaction pathways are marked with the following colors: purple, transporter, efflux pumps and secretion systems; orange, signal transduction and regulation mechanisms; green, polysaccharide degradation; red, competitive interactions; blue, bacteria-plant interaction pathways.

presented in all three bacteria, while biotin (B$_7$) pathway can be referred just to *Variovorax* (Fig. 4; Table S7).

**Cell-cell communication as a key component of microbe-alga interaction.** In the *Dyadobacter* genome the *darA* and *darB* genes, coding for the autoinducer synthase, were highly transcribed during exponential and stationary growth phases (6,376 and 12,094 hits). These genes are involved in the dialkylresorcinol synthesis gene cluster of *Dyadobacter*. Dialkylresorcinols are known as novel and widespread bacterial signaling molecules (23). Further analysis of the cell-cell communication as a key component of microbe-alga interaction identified two autoinducer synthase genes (i.e., *luxI* homologues) coding for *N*-acyl-L-homoserine lactone synthases in *Variovorax*.

Additional searches for QQ genes identified 19 genes that were mostly transcribed at the stationary growth phase. A potential dienelactone hydrolase from *Dyadobacter* revealed 1,110 counts and a *Porphyrobacter* homologue had 1,212 counts of mRNA reads.

Overall, among other components of cell signaling, we identified four *luxR* solo coding genes being highly expressed in *Dyadobacter* and having 4,429 counts. Similarly, in *Variovorax*, three *luxR* genes were found with 2,761 mRNA read counts during the stationary growth phase. A similar pattern was observed in the exponentially growing cultures (Fig. 4; Table S7).

**Secretion systems are most relevant for competitive interaction.** The analysis revealed secretion systems, which are known to be beneficial for bacterial surface colonization and for the algal-bacterial interactions (24, 25). T4SS, T6SS, and T9SS were expressed by the different bacteria. A more detailed inspection of the genomes revealed that *Dyadobacter* coded for T5SS, T6SS, and T9SS, whereas the genomes of *Porphyrobacter* and *Variovorax* coded for T2SS, T4SS, T5SS, and T6SS. Genes affiliated with T1SS and T3SS were not found during this analysis.

The T9SS was strongly expressed by *Dyadobacter*, with 12,967 hits in the exponential and 58,578 hits in the stationary growth phases. Other components of transport and secretion systems were upregulated during the stationary growth phase, often with gene hits that were twice as high. In our data set, *Variovorax* was the only bacterium that expressed the T2SS. While counting 1,843 hits during the exponential growth phase, this number gets doubled during the stationary phase. Gene counts for the T4SS found in *Porphyrobacter* during the exponential growth phase increased by a factor of 7 in the stationary phase.

***Dyadobacter* expresses many polysaccharide-modifying enzymes.** The Bacteroidetes phylum is known for its ability to degrade a wide range of polysaccharides, a trait that has enabled its dominance in many diverse environments, including plants and algal polysaccharides (26). As a member of this phylum, HH091, codes for a wealth of

polysaccharide-modifying enzymes, many of which were highly expressed (Table 1; Fig. 4). $\alpha$-L-Fucosidases of GH29 and GH95 families were observed in the metatranscriptome. Highly transcribed genes also included a $\beta$-fructosidase, which belongs to the GH32 family. Additionally, genes predicted to be involved in chitin, xylan, xyloglucan, and galactoglucomannan utilization were highly expressed. Since HH091 was the most active organism to encode polysaccharide utilization loci (PULs), it revealed a remarkably high-level transcription of these genes. PULs of HH091 were expressed with 253,760 RNA-Seq reads at exponential and 486,271 at the stationary growth phase (Fig. 4).

**Mutualism and commensalism as the main survival concepts in phycosphere.** Fig. 4 reflects the expression of genes affiliated with competitive interaction of the microbiota. Genes that are responsible for competitive interaction pathways included potential antibiotic substance biosynthesis pathways, nucleases, permeases, proteases, and genes involved in heme biosynthesis. In addition, 128,759 hits were related to genes linked to reactive oxygen species (ROS) tolerance in the HH091 data set. During stationary growth phase, 89,165 hits versus 39,594 hits of exponential growth phase were observed (Fig. 4). The occurrence and expression of ROS tolerance genes in bacteria are essential for resistance to oxidative stress caused by the host (27).

The expression of invasion-associated proteins during the exponential growth phase is mostly negligible but gains importance during the stationary growth phase, as observed for *Variovorax* and *Dyadobacter* (Fig. 4). Further, we observed a relatively high number of hits for leucine-rich repeat (LRR) and invasion-associated proteins belonging to *Dyadobacter* and *Variovorax*. It is known that bacteria use LRR and invasion proteins as signaling and detecting components for the establishment of the interaction with the plant innate immune system (28).

**Phytohormones biosynthesis systems essential for plant-bacteria liaison.** Phytohormones play a vital role in plant growth and development as a regulator of numerous biological processes that can positively influence the growth and development of microalgae (8). To determine the involvement of the microalgae-growth-promoting substances, we have analyzed the enzymatic systems responsible for phytohormones biosynthesis. While *Dyadobacter* and *Porphyrobacter* appear to lack any genes affiliated with plant hormone production, the *Variovorax* genome codes for two enzymatic systems, nitrilase and nitrile hydratase/amidase, that are predicted to convert indole-3-acetonitrile to the plant hormone indole-3-acetic acid. *Variovorax* lacks tryptophan 2-monooxygenase, although it can produce indole-3-acetic acid using indole-3-acetonitrile as the precursor. In the transcriptome data, a total of 9,915 counts was linked to this pathway, implying that auxins are potentially produced and released. These data imply that *Variovorax* produces auxin-like molecules, possibly stimulating the growth of the microalga. Taken together, these data imply that each of the three bacteria transcribes a unique set of genes that are of relevance for synthesis of common goods and growth and survival of the whole community.

## DISCUSSION

A comprehensive understanding of the composition of the microbial community, as well as competitive interaction, is required to create scientific and theoretical fundamentals of interaction mechanisms between microalgae and other microorganisms, including the development of effective processes for simultaneous algal cultivation with enhancing the efficiency of microalgae biomass growth and associated valuable compounds production.

**Specialized distribution of assignments at algae-bacterial phycosphere.** The evaluation of the top 250 gene hits resulted in the identification of genes related to the general metabolic activities, carbohydrate degradation, biofilm formation, transport mechanisms, and secretion systems, which support the distribution of assignments at the microbial consortium. Former reports on the relationships within the microalga-bacteria consortia have also provided a blueprint for the construction of mutually beneficial synthetic ecosystems, in which the general metabolic activities played a significant role (29–31).

Many pathogenic bacteria are known to use secretion systems for the facilitation of their proliferation and survival inside eukaryotic hosts, typically by the secretion of protein effectors or protein-DNA complexes (32). The presence of T6SS suggests the distribution of tasks among members of studied consortia, providing the fitness and colonization advantages, which are not restricted to virulence. T9SS machinery components, established for *Dyadobacter* within the Bacteroidota phylum, represents the assembly of the gliding motility apparatus and possible external release of proteins with various functions, including cell surface exposition, attachment, and other virulence factors (33, 34). The widespread occurrence of gliding motility genes was previously revealed among the members of the same phylum, the gliding bacterium *Flavobacterium johnsoniae*, and the nonmotile oral pathogen *Porphyromonas gingivalis*. *F. johnsoniae* uses T9SS as the gliding motility apparatus and for the secretion of a chitinase that is required for chitin digestion, and the *P. gingivalis* secretes through this system gingipain protease virulence factors. The same route likely secretes other polysaccharide-digesting enzymes produced by other members of the phylum. The mechanisms underlying the processes of gliding motility and cell surface machinery to utilize polysaccharides remain unclear (35, 36). Nevertheless, it is known that T9SS machinery secretes most of CAZymes, such as GHs, PLs, CEs, and accessory proteins (37, 38).

**Complex relationships through competition and synergy in algae-bacterial phycosphere biofilms.** Studying the interactions between members of alga-bacterial phycosphere, we investigated the hypothesis that dominant bacterial members possess the role of superior competitors. Members of the microbiome of *S. quadricauda* participate in the consortium niche in a competitive way that is reflected in a heat map with the correspondence to genes, affiliated with potential antibiotic substances, endonucleases and exonucleases, permeases, proteases, heme synthesis, and QQ (Fig. 4), which are known as important factors required for biofilm formation, virulence, and competition (39–42). The analysis of the microbiome of *S. Quadricauda* revealed different proteins supposed to be beneficial during competition for space and nutrients on surfaces in biofilms. Bacterial dominance can be attributed to the ability of these organisms to rapidly form microcolonies and their ability to produce extracellular antibacterial compounds (43). The *S. quadricauda* microbiome is composed of single-species populations or mixed populations with various levels of interaction, depending on the exponential or stationary growth phase. *Porphyrobacter*, *Dyadobacter*, and *Variovorax* were found to be the dominant producers of numerous antibacterial proteins, which can possibly eliminate other microorganisms or exhibit strong inhibitory activity against them.

The signaling molecules related to QS and QQ activity were affiliated with the Alphaproteobacteriota and Bacteroidota. Among metatranscriptome data sets proteins predicted as QQ included dienelactone hydrolase, imidazolonepropionase, 6-phosphogluconolactonase, gluconolactonase, oxidoreductases, and metal-dependent hydrolases of the *β*-lactamase superfamily, related to QQ activity. Highly transcribed genes were observed, mostly at the stationary growth phase, that fulfill the competitive needs of bacteria to comprise one of the dominant heterotrophic bacterial groups in aquaculture, which are represented in Fig. 4. Dienelactone hydrolase, known as a QQ enzyme that degrades or modifies *N-Acyl* homoserine lactones (AHLs) (44, 45), was established for *Dyadobacter* and *Porphyrobacter*. Gluconolactonases, reported as quorum-quenching enzymes (46), were mapped to *Dyadobacter*. Another class of enzyme, oxidoreductase, established to catalyze the oxidation or reduction of acyl side chain (33, 34), originated in *Variovorax* and *Dyadobacter*. The analysis also revealed several phenotypes beneficial for bacterial surface colonization, including motility, exopolysaccharide production, biofilm formation, and toxin production (Fig. 4), which are often regulated by QS (8, 16, 17).

Simultaneously, several members of the *S. quadricauda* microbiome appeared to be the main suppliers of vitamins to microalga. Genes involved in thiamin, cobalamin, biotin, and riboflavin synthesis were established for Alphaproteobacteriota and Cytophagaceae, which confirms the strong evidence that alga-associated bacteria are responsible for the supply of the essential vitamins to alga (19). Thus, our study shows that this interaction involves the strong collaboration between members of the alga-bacterial phycosphere

with the support of nutritive components and the synergetic exchange of biosynthetic compounds.

A significant number of genes of high importance for root colonization, biofilm formation, invasion (47, 48), virulence, and pathogenicity were identified (49–51). Fig. 4 describes genes affiliated with overall plant-bacteria interaction pathways, including ROS tolerance, LRR proteins, and invasion-associated proteins. Numerous genes responsible for the ROS tolerance were highly transcribed at the stationary phase, which explains the necessity of bacteria to protect itself from massive amounts of reactive oxygen species released by microalgae, which was previously suggested to expose them to pathogens (28). It is supposed that dominating microorganisms can use LRR and invasion proteins as a signaling and detecting components for the establishment of the interaction with the possible innate immune system of the *S. quadricauda*.

In summary, the current study gives a detailed insight into mutualistic collaboration of microalgae and bacteria, including the involvement of competitive interplay between bacteria. Future work will now have to unravel the signaling between the bacteria and eukaryotes, as well as the detailed nutrient exchange and mutual support in different aspects of cross-kingdom synergistic network.

## MATERIALS AND METHODS

**Microorganisms used in this study and cultivation media.** *S. quadricauda* MZCH 10104 was obtained from the Microalgae and Zygnematophyceae Collection Hamburg (MZCH) and cultivated in BG11 medium at $20 \pm 1°C$ and $100 \pm 10$ $\mu$mol photons m$^{-2}$ s$^{-1}$ with a 14-h light/10-h dark schedule (19, 52, 53). To maintain the axenity of the algal culture, *S. quadricauda* was treated with the antibiotic cocktail: penicillin G, streptomycin sulfate, and gentamicin sulfate (100, 25, and 25 mg/L, respectively) (54–56).

The media for cultivation of individual bacterial isolates derived from the microalgae-associated community were prepared as follows. R2A medium and TYES were prepared as described previously (57, 58), and M9, TSB, and NB media were prepared according to the method of Sambrook and Russell (59). To stimulate microbial growth, the media were in part supplemented with algal culture extracts during the exponential and stationary growth phase of the microalgae ranging from 5 to 50% (vol/vol). The inoculated plates were incubated for 5 to 7 days at 22°C under aerobic and anaerobic conditions.

*Dyadobacter* sp. HH091 was isolated during this work from a laboratory culture of *S. quadricauda* MZCH 10104. The isolate was routinely grown in 5 mL of tryptone yeast extract salts (TYES) broth at 22°C for 3 to 4 days at 200 rpm (60).

**Coculturing procedure and conditions.** *S. quadricauda* MZCH 10104 and *Dyadobacter* sp. HH091 were cocultured in BG11 medium at $20 \pm 1°C$ and $100 \pm 10$ $\mu$mol photons m$^{-2}$ s$^{-1}$ with a 14-h light/10-h dark schedule over a time period of 13 days. Therefore, 1 mL of *S. quadricauda* was treated with an antibiotic cocktail of penicillin$_{100}$, streptomycin$_{25}$, and gentamycin$_{25}$ in 50 mL of BG11 medium to remove all bacteria. The antibiotic treatment was performed for 1 day. Afterwards, the microalga was centrifuged (5,000 rpm for 10 min) and washed two times with 1 mL BG11 and finally resuspended in 50 mL of BG11 medium, where it was grown for 13 days. At the start of the experiment, each flask contained 50 mL of BG11, *S. quadricauda* (optical density at 750 nm [OD$_{750nm}$] = 0.03), and *Dyadobacter* (OD$_{600nm}$ = 0.05).

**Bacterial RNA extraction and sequencing.** The hot phenol method with minor modifications was used to extract the total RNA (19, 61). RNA quality was checked using a 2100 Bioanalyzer with the RNA 6000 Nano kit (Agilent Technologies). The RNA integrity number (RIN) for all samples was ≥7. Equal amounts of the remaining transcripts and kit components were used for cDNA library construction. Libraries suitable for sequencing were prepared from 400 and 275 ng of total RNA with oligo(dT) capture beads for poly(A) mRNA enrichment using the TruSeq stranded mRNA library preparation kit (Illumina) according to the manufacturer's instructions. After 14 cycles of PCR amplification, the size distribution of the barcoded DNA libraries was estimated ~300 bp by electrophoresis on Agilent DNA HS bioanalyzer microfluidic chips.

Sequencing of pooled libraries spiked with 5% PhiX control library was performed at 8 million reads/sample in paired-end mode with 150-nucleotide (nt) read length on the NextSeq 500 platform (Illumina) using a High Output 400M sequencing kit. Demultiplexed FASTQ files were generated with bcl2fastq2 v2.20.0.422 (Illumina).

**Processing and analysis of RNA-Seq reads.** Fastp (v0.21.0) was used to remove artificial and low-quality (Phred quality score below 15) sequences from the 3'-end of sequence reads (62). Putative base calling errors located in regions were two reads of a read pair overlap were corrected (option: –correction). Kraken2 (v2.1.2) was used in combination with Bracken (v2.6.2) to assess the taxonomic composition (63, 64). Additionally, reads were assembled with Trinity (v2.13.2) (65). Assembly statistics were assessed with Quast (v5.0.2) (66). The resulting contigs were aligned to sequences present in the UniProtKB/Swiss-Prot database (release 2021_04) and taxonomically annotated accordingly (67). Both were achieved with Mmseqs2 (version ad5837b3444728411e6c90f8c6ba9370f665c443) in "easy taxonomy" mode (–lca-mode 4) (68).

**Total bacterial DNA extraction.** Genomic DNA of pure cultures of the strain *Dyadobacter* sp. HH091 was extracted using the peqGOLD bacterial DNA kit (PEQLAB Biotechnologie GmbH, Erlangen, Germany) according to the manufacturer's instructions.

**Dyadobacter sp. HH091 transformation.** The strain HH091 was transformed with modified plasmid pBBR1MCS-5-eGFP by electroporation according to standard methods, which resulted in bright green fluorescent colonies as observed by fluorescence microscopy (59). The plasmid contains the broad-host-range vector pBBR1MCS-5, providing a gentamicin resistance and the expression of GFP. Gentamycin was provided at 100 $\mu$g/mL, and the bacteria were grown for 3 to 4 days at 22°C in liquid medium or on a plate.

**Bacterial genome sequencing, *de novo* assembly, and binning.** Total genomic DNA of *Dyadobacter* sp. HH091 was extracted for a genomic analysis using the NucleoBond high-molecular-weight genomic DNA kit for microorganisms (Macherey-Nagel, Germany) following the manufacturer's instructions and a previously published enzymatic cell lysis protocol with some modifications, including freezing in liquid nitrogen, bead beating, and an additional lysis pretreatment with proteinase K and lysozyme for 24 h at 55°C (2). The extracted DNA was sequenced on an Illumina NextSeq 500 platform using rapid sequencing by synthesis (SBS) chemistry v2 (Illumina, San Diego). For this, the DNA library was constructed applying the NEBNext Ultra II DNA library prep kit for Illumina (New England Biolabs) according to the manufacturer's protocol. Initial fragmentation of DNA was performed on the Bioruptor NGS (Diagenode) with 30 s on/30 s off for 16 cycles. Sequencing of the metagenomic DNA library was performed on the NextSeq 500 platform (Illumina) as paired-end run ($2 \times 150$ cycles) with ~60 mio reads. Fastp (v0.21.0) was used to remove artificial and low-quality (Phred quality score below 15) sequences from the 3'-end of sequence reads. Putative base calling errors located in regions were two reads of a read pair overlap were corrected (Fastp option: –correction). Reads shorter than 40 bp were discarded. Sequence reads were than assembled using SPAdes (v3.15.3) (69). The final draft assembly consists of 7,862,706 bp with a GC content of 43.81%. The final genome assembly resulted in 2,018 contigs (N50, 607,803 bp; L50, 10), with a largest contig size of 1,195,963 bp.

**Physiological analyses. (i) Microscopy investigations.** *Dyadobacter* sp. HH091 expressing eGFP was cocultured with *S. quadricauda* MZCH 10104 and studied using a confocal laser scanning microscope (CLSM) Axio Observer.Z1/7 LSM 800 (Carl Zeiss Microscopy GmbH, Jena, Germany), including Z-Stack microscope techniques. The analysis of the CLSM images were done with ZEN software (version 2.3; Carl Zeiss Microscopy GmbH). An improved 4'-6-diamidino-2-phenylindole (DAPI) staining procedure with some modifications was used in microscopy investigations (70). Modifications included the treatment with the TrueVIEW autofluorescence quenching kit (Vector Labs, SP-8400), which was employed to enhance staining and to lower the autofluorescence of chlorophyll of the microalga known to be troublesome. Background autofluorescence occurring in the 600- to 700-nm range makes it impossible to detect the bacteria transformed with plasmids expressing fluorescent proteins. The TrueVIEW Quencher is an aqueous solution of a hydrophilic molecule, which binds to chlorophyll electrostatically and lowers the fluorescence (71).

**(ii) Pulse-amplitude-modulation (PAM) fluorometry.** The photosynthetic activity of microalgae was measured by pulse-amplitude-modulation (PAM) fluorometry. The measured parameters represent the optimal quantum yield of PS II photochemistry ($F_v/F_m$), with the fluorescence ($F_0$) measured during the illumination of a pre-dark-adapted sample with open reaction centers and under saturating light with closed reaction centers is the maximum fluorescence ($F_m$). The difference $F_m - F_0$, is called variable fluorescence ($F_v$), representing the amount of light energy that can be used by PS II (72).

**(iii) Fluorescence-activated cell sorting (FACS).** Flow cytometry was applied to analyze the chlorophyll content of *S. quadricauda* cocultivated with *Dyadobacter* sp. HH091. An antibiotic-treated algae culture without *Dyadobacter* sp. served as a control. Growth of *S. quadricauda* was monitored over 13 days after inoculation (start $OD_{750nm} = 0.090$) with and without *Dyadobacter* sp. ($OD_{600nm} = 0.05$). Culture samples with volumes of 1 mL (with and without *Dyadobacter*) were withdrawn in triplicate for the experiments every 3 days. For every measurement, we used 0.5 mL of algal culture diluted in 0.5 mL of BG11 medium and filtered through a 35-$\mu$m Strainer cap. The samples were subjected to flow cytometry using the S3e cell sorter (Bio-Rad, Hercules, CA) equipped with a 488-nm excitation laser and detectors for side and forward scatter (i.e., SSC and FSC area), FL1 (525/30-nm band pass filter), and FL2 (560-nm long-pass filter). Data analysis was carried out with the FlowJo software package (v10.6.1; BD Life Science, Ashlan, OR). The FL2 detector allowed detection of the chlorophyll autofluorescence of *S. quadricauda*, and the culture samples were analyzed in a two-dimensional density plot of FL2 area versus FSC area to identify algal cells with elevated and reduced chlorophyll levels, and bacteria to determine the relative ratio of each population. The identity (algae or bacteria) and physiological state (live or dead algae) of the populations, detected in the FL2-FCS area plot, was confirmed by microscopy after sorting the respective subpopulations with the cell sorting function (purity mode) of the cell sorter instrument. For each sample the measurement was normalized until 15,000 events.

**Data availability.** Sequences obtained for this study were submitted to the European Nucleotide Archive (ENA). They are publicly available under accession number PRJEB23338. Assembly of the *Dyadobacter* sp. HH091 genome is available via IMG/M (https://img.jgi.doe.gov) using IMG ID 2842103827.

## SUPPLEMENTAL MATERIAL

Supplemental material is available online only.

**SUPPLEMENTAL FILE 1**, PDF file, 2.7 MB.

## ACKNOWLEDGMENTS

We thank the members of the Microalgae and Zygnematophyceae Collection Hamburg (MZCH) for helpful discussion and for providing the microalgae sample. We thank the Core Unit SysMed at the University of Würzburg for excellent technical support, RNA-Seq data generation, and analysis.

This work was in part supported by the Deutscher Akademischer Austauschdienst (German Academic Exchange Service, DAAD), the Federal Ministry of Education and Research (Bundesministerium für Bildung und Forschung, BMBF) projects: MarBioTech (FKZ 031A565) and AquaHealth (FKZ 031B0945C), and the Center for Clinical Research (Interdisziplinäres Zentrum für Klinische Forschung, IZKF) project Z-6.

We declare no conflict of interest.

Y.A. and I.K. contributed to experimental design; lab work of phylogenetic, metatranscriptomic, genomic, bioinformatic, and physiological analytical approaches; and writing of the research article. D.I. and L.B. contributed to lab work of the genomic approaches. M.G. and E.K. contributed to lab work of metatranscriptomic approaches. M.A. and M.Q. contributed to assembly of metatranscriptomic data sets and bioinformatics approaches of the genomic data set. M.G., S.R., and I.d.G. contributed to lab work of physiological analytical approaches. D.H., W.R.S., and I.K. contributed to general experimental design and writing of the research article. All authors contributed to manuscript revision, and all read and approved the submitted version.

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
