## [Reviewer comments · Microbiology Spectrum]

Microbiology Spectrum

Microalgae and bacteria interaction – Evidence for division of diligence in the alga microbiota

Yekaterina Astafyeva, Dieter Hanelt, Wolfgang Streit, Malik Alawi, Marno Gurschke, Imke de Grahl, Lutgardis Bergmann, Daniela Indenbirken, Sigrun Reumann, Elena Katzowitsch, Minyue Qi, and Ines KROHN

Corresponding Author(s): Ines KROHN, University of Hamburg, Institute of Plant Science and Microbiology

Review Timeline:

Submission Date:	February 18, 2022
Editorial Decision:	May 16, 2022
Revision Received:	June 8, 2022
Accepted:	July 7, 2022

Editor: Eva Sonnenschein

Reviewer(s): The reviewers have opted to remain anonymous.

Transaction Report:

DOI: <https://doi.org/10.1128/spectrum.00633-22>

May 16, 2022

Dr. Ines KROHN
University of Hamburg, Institute of Plant Science and Microbiology
Microbiology and Biotechnology
Ohnhorststr. 18
Hamburg 22609
Germany

Re: Spectrum00633-22 (Microalgae and bacteria interaction - Evidence for division of diligence in the alga microbiota)

Dear Dr. Ines KROHN:

Link Not Available

Sincerely,

Eva Sonnenschein

Journals Department
Reviewer comments:

Reviewer #1 (Comments for the Author):

Astafyeva et al., present data about the role of key bacteria in the phycosphere of the alga *Scenedesmus quadricauda*. Using lab experiments the authors determine that the genus *Dyadobacter* is central in promoting algal growth and photosynthetic activity. Using transcriptomics, the authors further investigate potential interactions among multiple bacteria and their algal host, and suggest that the metabolic exchange between the 3 dominant genera in the phycosphere- *Variovorax*, *Porphyrobacter* and *Dyadobacter*- includes specialized metabolites produced by each genus.

The algal-bacterial metabolic exchange is a timely and environmentally important topic. In my view, the novelty of this work is in

its ability to study both a co-cultures and a poly-microbial population. An algal host with several bacterial partners offers a model system with increased biological and ecological complexity, and that's valuable.

General comment

- There is a conceptual gap between the co-cultures experiments that point to an interesting influence of *Dyadobacter* on the alga, and the transcriptomics part where a non-axenic algal culture is analyzed. In the non-axenic culture multiple bacteria interact with each other and with the algal host. These are two very different systems and I found it difficult to connect between these two parts and draw conclusions from one system to another.
- There is no explicit presentation of the research question. What was the goal of the study? Line 117 presents the focus of the work as elucidating the interactions of *Dyadobacter* with other bacteria and with the algal host but it is not clear why specifically this bacterium is at the heart of this statement. And if indeed this bacterium is of interest, why not focus on it also in the transcriptomic part? Why shift to a general analysis of the entire bacterial community? It seems to me that the genetic tools that the authors developed for *Dyadobacter* make it an attractive candidate for further study. It could nicely be the focus of the experimental and transcriptomic part of this study. Importantly, many of the hypotheses that were generated by the transcriptomic work could be experimentally validated using *Dyadobacter* culture experiments.
- The manuscript requires major professional language editing.
- The author should carefully revise the references they use to support their claims. Some references have no relevance to the text (example: lines 247-248).

Specific comments

Line 96: "30 species" limitation is a very interesting fact and requires appropriate references.

Lines 97-98: Not sure this is accurate. In fact, pathogenic *Roseobacters* are often associated with microalgae but somehow appear to live in a balanced manner in the community. For example, *Dinoroseobacter* (Wang H, Tomasch J, Jarek M, Wagner-Döbler I. A dual-species co-cultivation system to study the interactions between *Roseobacters* and dinoflagellates. *Front Microbiol.* 2014) and *Phaeobacter* (Dynamic metabolic exchange governs a marine algal-bacterial interaction, Segev E., Wyche T. P., Kim K. H. et al. (2016) *eLife.*) are algal pathogens that are commonly found associated with phytoplankton in the environment.

Line 206: Authors should include in the Methods section the co-culturing procedure and conditions.

Lines 206-273: Can the authors briefly explain in the text why these genes represent an adaptation to life in a multispecies community? Why are these sets of genes "remarkable"?

Line 275: Do you think that the pBBR plasmid is burdening bacteria? Is there a difference in the growth curves of WT bacteria versus plasmid-expressing bacteria? These data should be included.

Line 279: The images in Figure 1 can't support a general statement like "always tightly associated". While an association is seen, there's no quantification of the planktonic bacteria, and there's no control image with no bacteria.

Line 283: I don't understand how the genomic information and successful transformation lead to this question.

Line 287: This difference is really nice! It would be valuable to also present growth curves of algae and bacteria.

Line 291: The raw data of flow cytometry in Figure 2C,D,E should be synthesized into an informative graph. Raw data can go to the supplemental.

Lines 298-299: The labels of Figure 2B are different in the text and in the figure.

Line 304: Figure 2E shows 3.24% bacteria in an axenic culture? So is the culture non-axenic?

Line 309: This is confusing; why not assess bacterial gene expression in a co-culture with algae? What's the link between the previous observations in co-culture and transcriptomics in a multi-species community?

Lines 311-315: To compare between the experimental and transcriptomics part, authors should at least present growth curves of axenic versus non-axenic cultures. The authors refer to "exponential and stationary growth phases" so the data has to be shown. Do algae perform better in the presence of bacteria in the non-axenic culture? Is this comparable to the co-cultures with *Dyadobacter*?

Line 334: If this non-axenic culture contains multiple bacteria, and the transcriptomic data can only reach the class level, how can the authors assign transcripts to a specific genus? Also- were transcripts levels normalized to bacterial cell numbers? This information is missing.

Line 385: Why do the authors conclude that these genera are B12 auxotrophs? The entire section about B vitamins is very interesting and has great potential, but is highly speculative at this point. Simple experiments could determine which bacteria are auxotrophs.

Lines 387-389: According to Figure S7 the aerobic B12 biosynthetic genes are quite similar between *Dyadobacter* and *Variovorax*. Why do the authors state that '*Variovorax* codes in its genome an anaerobic pathway of the vitamin B12 biosynthesis. Thus, it is likely that vitamin B12 was provided by *Variovorax* and not by any of the two other bacteria; as they lack these genes'? Furthermore, according to Figure 4, B12 biosynthesis genes are highly expressed in *Porphyrobacter* and *Dyadobacter*. This section requires clarification.

Lines 411-437: Among the highly expressed genes in *Dyadobacter*, which are expected to be involved in the beneficial

interaction with the algal host? In what manner?

Figure 1: What is "P"? What are the different panels in 1C?

Reviewer #2 (Comments for the Author):

Astafyeva et al investigated the holobiont of the alga *Scenedesmus quadricauda* and characterized the facultative associated bacteria using molecular taxonomy. The authors further provide a comparative transcriptomics study of the host grown axenic or with the bacteria, hereby deriving information on the potential chemical exchanges in the alga-bacteria interactions. As only one bacteria was cultivable, the referent genomes available from the uncultivable species were used to assign the active genes in the transcriptomics analysis.

The main outcome is the identification of the shared currency between alga and bacteria which involved vitamins and microalgae-growth promoting factors. The dominant bacteria identified as *Variovorax*, *Porphyrobacter* and *Dyadobacter* produced respectively auxins, B-vitamins and polysaccharides degradation enzymes. It is postulated that these bacteria are competing based on the genes identification, the study could be supported with an in vitro competition experiments, but the authors could not isolate two of the bacteria from their strain in cultures. The originality of the paper resides in finding that *Scenedesmus quadricauda* and its associated symbiont share common metabolic patterns than for other algal holobionts. The results have been deeply unravelled to enable the proposition of a number of hypothesis on the activities and function of the bacteria associated with the algal holobiont. A metabolome profiling using a complementary omics method could strengthen the transcriptomics results with the identification of secreted metabolites in the cocultures.

The primary outcome message delivered relatively clearly and quality controls of the method and the results are strongly supported. The discussion is also very well elaborated and the authors clearly exposed their hypothesis relative to the observations. One caveat is that the sequences could not be found under the accession number PRJEB23338 at <https://www.ebi.ac.uk/ena/browser/view/PRJEB23338>. Please ensure the data is publicly available at the time of publication.

I summarized more specific suggestions to help improve the manuscript in the following.

Line 91 consider remove the "of" before bacteria

Line 100 remove the word "special"

Line 103 remove the words "in general"

Line 107 to 112 Add a reference to these statements

Line 131. Can the strain HH091 be deposited in public collection?

Line 137 spell out what RIN means

Line 140 what does mean (1/2 volume), please clarify

Line 199-201 at the time of this review, the genome and sequences could not be found under the numbers given, please ensure the data is easy and publicly available.

Line 213 clarify how the autofluorescence of chlorophyll was lowered and what specific modifications does the kit TrueView

Line 227 clarify if the algae culture without *Dyadobacter* was axenic or not

Line 230 how much volume was withdrawn?

Line 229 how much volume and culture served for inoculation?

Please add the protocol of sample preparation used for fixation of the cells and get them ready for flow cytometry analysis

Line 247, from the first statement in results, it is unclear whether it is concerning the alga used in the study, please rephrase

Line 252. It is explained individual strains were isolated from the alga, but it is not mentioned in the methods how this was done.

Please add the information line 131 (was it done by cells or supernatant of culture, what inoculation volume, age of the algal culture etc)

Line 267 remove the ' from its

Line 279 further describe if it is attached to the cells, or the EPS matrix coat.

Line 277 please refer what is the abbreviation CSML earlier in the text

Figure 1: explain what means P letter in the caption. Is the bacteria inside inside the cells? Is it unclear whether the green fluorescence seen belong to the bacteria or the algal chloroplasts, I can recommend to use another arbitrary color to distinguish

the two fluorescences, or show the pictures without the autofluorescence of the chlorophyll a
Line 288. Visible after 1-2 days. Why not always at the same time point, did you observe differences in the replicates?
Line 383 consider use drivers instead of drives
Lines 497-500 add a reference to this statement

Staff Comments:

Preparing Revision Guidelines

Please return the manuscript within 60 days; if you cannot complete the modification within this time period, please contact me. If you do not wish to modify the manuscript and prefer to submit it to another journal, please notify me of your decision immediately so that the manuscript may be formally withdrawn from consideration by Microbiology Spectrum.

UHH – Institute of Plant Science and Microbiology Ohnhorststr. 18, 22609 Hamburg

Dr. Ines Krohn

Biozentrum Klein Flottbek
Department Biologie
Abteilung für Mikrobiologie & Biotechnologie
Ohnhorststraße 18
Raum 3.156
22609 Hamburg

Tel. +49 40 42816 444

Fax +49 40 42816 459

ines.krohn@uni-hamburg.de

Response to reviewer

07.07.22

Dear Editors and reviewer,

Please find attached the revised version of our manuscript “Microalgae and bacteria interaction – Evidence for division of diligence in the alga microbiota” for publication in ASM Microbiology Spectrum. We would like to thank you very much for your comments. We have integrated them into the current version and made some additions marked in green. However, we think this manuscript fits well into the journal, because it describes a fundamental principle with respect to the key features of possible competitive and plant-bacteria interaction pathways of microalga-bacteria communities.

Reviewer comments:

Reviewer #1 (Comments for the Author):

Astafyeva et al., present data about the role of key bacteria in the phycosphere of the alga *Scenedesmus quadricauda*. Using lab experiments the authors determine that the genus *Dyadobacter* is central in promoting algal growth and photosynthetic activity. Using transcriptomics, the authors further investigate potential interactions among multiple bacteria and their algal host, and suggest that the metabolic exchange between the 3 dominant genera in the phycosphere- *Variovorax*, *Porphyrobacter* and *Dyadobacter*- includes specialized metabolites produced by each genus.

Comment: Thank you for your comment.

The algal-bacterial metabolic exchange is a timely and environmentally important topic. In my view, the novelty of this work is in its ability to study both a co-cultures and a poly-microbial population. An algal host with several bacterial partners offers a model system with increased biological and ecological complexity, and that's valuable.

Comment: Thank you.

General comment

- There is a conceptual gap between the co-cultures experiments that point to an interesting influence of *Dyadobacter* on the alga, and the transcriptomics part where a non-axenic algal culture is analyzed. In the non-axenic culture multiple bacteria interact with each other and with the algal host. These are two very different systems and I found it difficult to connect between these two parts and draw conclusions from one system to another.

Comment: Thank you. This is a very interesting and relevant point to discuss. The transcriptomics of an axenic algal culture was not analyzed during that period of time. We already had some problems during the cultivation by using an axenic microalga, so we decided to use an antibiotic treatment for our experiments to minimize the bacterial colonization. Please see line 151. We hypothesize that more than one compound is essential for a healthy algae growth, which is not known yet. Please see line 351. To figure out the importance of the *Dyadobacter* strain, we used the antibiotic-treated microalgae culture as a control for the following experiments. All experiments were done in triplicates.

The co-culture of *Dyadobacter* and *S. quadricauda* was examined in comparison to control antibiotic-treated microalgae using confocal microscopy and PAM fluorometry (photosynthetic activity measurements). Additional FACs analyses were implemented to verify that the number of viable and photosynthetic active cells in the presence of *Dyadobacter* was much higher in comparison to antibiotic-treated cultures (lines 323-347).

- There is no explicit presentation of the research question. What was the goal of the study? Line 117 presents the focus of the work as elucidating the interactions of *Dyadobacter* with other bacteria and with the algal host but it is not clear why specifically this bacterium is at the heart of this statement.

Comment: Thank you for your comment. We added the information about the goal of this study to the revised manuscript. Please see lines 118-123.

And if indeed this bacterium is of interest, why not focus on it also in the transcriptomic part? Why shift to a general analysis of the entire bacterial community? It seems to me that the genetic tools that the authors developed for *Dyadobacter* make it an attractive candidate for further study. It could nicely be the focus of the experimental and transcriptomic part of this study. Importantly, many of the hypotheses that were generated by the transcriptomic work could be experimentally validated using *Dyadobacter* culture experiments.

Comment: Thank you for your comment. We were able to cultivate a single *Dyadobacter* affiliated with a *S. quadricauda* laboratory non-axenic culture. The isolation of *Porphyrobacter* and *Variovorax* from the same *S. quadricauda* laboratory non-axenic culture was unsuccessful (lines 287). We rephrase this part, to answer your question in line 118 onwards.

- The manuscript requires major professional language editing.

Comment: Thank you, we corrected the grammatical issues. Please let us know, if we overlooked something.

- The author should carefully revise the references they use to support their claims. Some references have no relevance to the text (example: lines 247-248).

Comment: Thank you for your comment. We have corrected it (line 279)

Specific comments

Comment: Thank you so much.

Line 96: "30 species" limitation is a very interesting fact and requires appropriate references.

Comment: We have corrected it and added the relevant references. Please see the line 97.

Lines 97-98: Not sure this is accurate. In fact, pathogenic Roseobacters are often associated with microalgae but somehow appear to live in a balanced manner in the community. For example, *Dinoroseobacter* (Wang H, Tomasch J, Jarek M, Wagner-Döbler I. A dual-species co-cultivation system to study the interactions between Roseobacters and dinoflagellates. *Front Microbiol.* 2014) and *Phaeobacter* (Dynamic metabolic exchange governs a marine algal-bacterial interaction, Segev E., Wyche T. P., Kim K. H. et al. (2016) *eLife.*) are algal pathogens that are commonly found associated with phytoplankton in the environment.

Comment: We have rewritten this part according to your suggestions. Please see the line 98-101.

Line 206: Authors should include in the Methods section the co-culturing procedure and conditions.

Comment: Thank you for your comment. We added the co-culturing procedure and conditions in the Material and Methods section (line 147-157).

Lines 206-273: Can the authors briefly explain in the text why these genes represent an adaptation to life in a multispecies community? Why are these sets of genes "remarkable"?

Comment: Thank you for your questions. We have added a more detailed explanation. We find these sets of genes "remarkable", because they play an important role in competitive and plant-bacteria interaction. Please see the line 298-303.

Line 275: Do you think that the pBBR plasmid is burdening bacteria? Is there a difference in the growth curves of WT bacteria versus plasmid-expressing bacteria? These data should be included.

Comment: Thank you for your comment. This is an important fact. We have done growth experiments to fix it and we have added these data. Please see the lines 316-317 in the revised version.

Line 279: The images in Figure 1 can't support a general statement like "always tightly associated". While an association is seen, there's no quantification of the planktonic bacteria, and there's no control image with no bacteria.

Comment: Thank you for your comment. We have rewritten this part in a more general way. Please see the lines 319 in the revised version.

Line 283: I don't understand how the genomic information and successful transformation lead to this question.

Comment: We put this information to the place, where it suits better. Please check the line 312-316.

Line 287: This difference is really nice! It would be valuable to also present growth curves of algae and bacteria.

Comment: Sorry for this typo. We corrected this to 2-3 days. Please see the line 326. We have rewritten the part and added the growth curves of the microalgae and the bacteria. Please check lines 324-326. We summarized these data in FIGURE 2A.

Line 291: The raw data of flow cytometry in Figure 2C, D, E should be synthesized into an informative graph. Raw data can go to the supplemental.

Comment: Thank you for your comment. We totally agree and we have re-arranged FIGURE 2. FIGURE 2B summarized the comparison of the populations based on the chlorophyll intensity. The raw data are now in the supplemental material part. Please see FIGURE S2.

Lines 298-299: The labels of Figure 2B are different in the text and in the figure.

Comment: We corrected this issue. Please see FIGURE 2 and line 335-337 revised version

Line 304: Figure 2E shows 3.24% bacteria in an axenic culture? So is the culture non-axenic?

Comment: Yes, you are right. It is not an axenic culture. We chose to use an antibiotic-treated culture. We edit this in revised manuscript.

Line 309: This is confusing; why not assess bacterial gene expression in a co-culture with algae? What's the link between the previous observations in co-culture and transcriptomics in a multi-species community?

Comment: Thank you for this comment. We hypothesize that more than one compound is essential for a healthy algae growth, which is not known yet. We have implemented the multispecies analysis that included *Porphyrobacter* sp. AAP82 (IMG 2551306481), *Variovorax paradoxus* S110 (IMG 644736413), and *Dyadobacter* sp. HH091. To figure out the importance of the *Dyadobacter* strain, we performed the studies with it and used the antibiotic-treated microalgae culture as a control for the following experiments. We rephrased this part. Please see the lines 351-352.

Lines 311-315: To compare between the experimental and transcriptomics part, authors should at least present growth curves of axenic versus non-axenic cultures. The authors refer to "exponential and stationary growth phases" so the data has to be shown. Do algae perform better in the presence of bacteria in the non-axenic culture? Is this comparable to the co-cultures with *Dyadobacter*?

Comment: Thank you for this comment. We have added these data in the FIGURE 2. The performance of algae in the presence of bacteria is described in the lines 321-335. We compared the co-cultures with *Dyadobacter* with the antibiotic-treated microalgae culture.

Line 334: If this non-axenic culture contains multiple bacteria, and the transcriptomic data can only reach the class level, how can the authors assign transcripts to a specific genus? Also- were transcripts levels normalized to bacterial cell numbers? This information is missing.

Comment: Thank you. This is a very interesting point. We did not normalize the bacteria cell numbers to the transcripts. This was not the aim of this project so far. In this manuscript we focused on the general pathways for the associated bacteria. In general, we could count approximately 10-15 bacterial cells for one microalgae cell. This information is a result of our microscopic analyses. We added this information in the current version of the manuscript. Please see the line 342-343.

Line 385: Why do the authors conclude that these genera are B12 auxotrophs? The entire section about B vitamins is very interesting and has great potential, but is highly speculative at this point. Simple experiments could determine which bacteria are auxotrophs.

Comment: Thank you for this comment. We totally agree. We added this reference to the revised version. Please see the line 429.

Lines 387-389: According to Figure S7 the aerobic B12 biosynthetic genes are quite similar between *Dyadobacter* and *Variovorax*. Why do the authors state that '*Variovorax* codes in its genome an anaerobic pathway of the vitamin B12 biosynthesis? Thus, it is likely that vitamin B12 was provided by *Variovorax* and not by any of the two other bacteria; as they lack these genes'? Furthermore, according to Figure 4, B12 biosynthesis genes are highly expressed in *Porphyrobacter* and *Dyadobacter*. This section requires clarification.

Comment: This is very interesting comment. We added the information and the reference. Please see the lines 429-433.

Lines 411-437: Among the highly expressed genes in *Dyadobacter*, which are expected to be involved in the beneficial interaction with the algal host? In what manner?

Comment: Thank you. We have added the information. Please see the line 458-459 and 474-476.

Figure 1: What is "P"? What are the different panels in 1C?

Comment: We apologize for this typo. We added this information in the (FIGURE 1) of the revised version. P means *pyrenoid*. Please see the line 319-321 and FIGURE legend.

Reviewer #2 (Comments for the Author):

Astafyeva et al investigated the holobiont of the alga *Scenedesmus quadricauda* and characterized the facultative associated bacteria using molecular taxonomy. The authors further provide a comparative transcriptomics study of the host grown axenic or with the bacteria, hereby deriving information on the potential chemical exchanges in the alga-bacteria interactions. As only one bacteria was cultivable, the referent genomes available from the uncultivable species were used to assign the active genes in the transcriptomics analysis.

Comment: Thank you for this comment.

The main outcome is the identification of the shared currency between alga and bacteria which involved vitamins and microalgae-growth promoting factors. The dominant bacteria identified as *Variovorax*, *Porphyrobacter* and *Dyadobacter* produced respectively auxins, B-vitamins and polysaccharides degradation enzymes. It is postulated that these bacteria are competing based on the genes identification, the study could be supported with an in vitro competition experiments, but the authors could not isolated two of the bacteria from their strain in cultures. The originality of the paper resides in finding that *Scenedesmus quadricauda* and its associated symbiont share common metabolic patterns than for other algal holobionts. The results have been deeply unravelled to enable the proposition of a number of hypothesis on the activities and function of the bacteria associated with the algal holobiont. A metabolome profiling using a complementary omics method could strength the transcriptomics results with the identification of secreted metabolites in the cocultures.

Comment: Thank you for your comment. Metabolome would be a very interesting task, which offers new hints for secretion studies. Unfortunately, our results shows just a very low concentration of proteins in the supernatant, which was more or less detectable as hypothetical or flagellin related proteins. More research is needed to detect significant. This would be a part of our upcoming research and publications, after optimization of the protocols.

The primary outcome message delivered relatively clearly and quality controls of the method and the results are strongly supported. The discussion is also very well elaborated and the authors clearly exposed their hypothesis relative to the observations. One caveat is that the sequences could not be found under the accession number PRJEB23338 at <https://www.ebi.ac.uk/ena/browser/view/PRJEB23338>. Please ensure the data is publicly available at the time of publication.

Comment: Thank you. The datasets are now available under the mention accession number.

I summarized more specific suggestions to help improve the manuscript in the following.

Comment: many, many thanks.

Line 91 consider remove the "of" before bacteria.

Comment: We corrected this. Please see the line 92 in the revised version.

Line 100 remove the word "special"

Comment: We corrected this. Please see the line 102 in the revised version.

Line 103 remove the words "in general"

Comment: We corrected this. Please see the line 105 in the revised version.

Line 107 to 112 Add a reference to these statements

Comment: Thank you for your comment. We added the missing references. Please see the line 114.

Line 131. Can the strain HH091 be deposited in public collection?

Comment: Thank you for the question. So far, our strain *Dyadobacter* sp. HH 091 is not deposited in any official culture collection like DSMZ. We can make it available for you or other researchers regarding the "Material transfer agreements" of the University of Hamburg, Section of Microbiology and Biotechnology. Please let us know, if you are interested in any collaborations.

Line 137 spell out what RIN means

Comment: We added this information. Please see the line 162.

Line 140 what does mean (1/2 volume), please clarify

Comment: We apologize for this misunderstanding. We used equal amounts of the remaining transcripts and kit components were used for cDNA library construction. We have rewritten this part. Please see the lines 163-167.

Line 199-201 at the time of this review, the genome and sequences could not be found under the numbers given, please ensure the data is easy and publicly available.

Comment: The datasets are now available under the mentioned accession numbers.

Line 213 clarify how the autofluorescence of chlorophyll was lowered and what specific modifications does the kit TrueView

Comment: Thank you for your comment. We added a few sentences to the manuscript. Please see the line 239-242.

Line 227 clarify if the algae culture without *Dyadobacter* was axenic or not

Comment: We rephrased this part. Please see the lines 256-257.

Line 230 how much volume was withdrawn?

Comment: Thank for your question. Culture samples in the volume of 1 mL (with and without *Dyadobacter*) were with-drawn in triplicates for the experiments every three days. We add this information to the manuscript. Please see the line 259-262.

Line 229 how much volume and culture served for inoculation?

Comment: Thank you for your question. We added this information in the Material and methods section. Please see lines 148-157.

Please add the protocol of sample preparation used for fixation of the cells and get them ready for flow cytometry analysis

Comment: Thank you for the comment. For every measurement we used 0.5 mL of algal culture diluted in 0.5 mL of BG11 medium and filtered through a 35µm Strainer cap. Please see lines 261-262.

Line 247, from the first statement in results, it is unclear whether it is concerning the alga used in the study, please re-phrase

Comment: In this research we concentrated on the microalgae *Scenedesmus quadricauda* and its bacterial community. We rephrased this part. Please see line 279-281.

Line 252. It is explained individual strains were isolated from the alga, but it is not mentioned in the methods how this was done. Please add the information line 131 (was it done by cells or supernatant of culture, what inoculation volume, age of the algal culture etc.)

Comment: Thank you for your comment. We added this part in the material and methods section. Please see the line 135 – 142 in the revised version.

Line 267 remove the ' from its

Comment: We correct this typo. Please see line 300

Line 279 further describe if it is attached to the cells, or the EPS matrix coat.

Comment: We added this information to the current version of the manuscript. Please see the line 315.

Line 277 please refer what is the abbreviation CSML earlier in the text

Comment: We apologize for this typo. It is read CLSM. Please see line 318.

Figure 1: explain what means P letter in the caption.

Comment: We apologize for this typo. We added this information in the (FIGURE 1) of the revised version. P means *pyrenoid*.

Is the bacteria inside inside the cells?

Comment: Thank you for the question. We also added this information. Please see line 315.

Is it unclear whether the green fluorescence seen belong to the bacteria or the algal chloroplasts, I can recommend to use another arbitrary color to distinguish the two fluorescences, or show the pictures without the autofluorescence of the chlorophyll a

Comment: Thank you for your comment. Modifications included the treatment with TrueVIEW Autofluorescence Quenching Kit (Vector Labs, SP-8400), which was employed to enhance staining and to lower the autofluorescence of chlorophyll of the microalga known to be troublesome. Background autofluorescence occurring in the 600–700 nm range, makes it impossible to detect the bacteria transformed with plasmids expressing fluorescent proteins. The TrueVIEW Quencher is an aqueous solution of a hydrophilic molecule, which binds to chlorophyll electrostatically and lowers the fluorescence³⁹.

We added this information. Please see the lines 236-242.

Line 288. Visible after 1-2 days. Why not always at the same time point, did you observe differences in the replicates?

Comment: Sorry for this typo. We corrected this to 2-3 days. Please see the line 326. We have rewritten the sentence. The differences in all replicated started to be visible after second day and it continued in the same way after.

Line 383 consider use drivers instead of drives

Comment: Thank you. We replaced these words regarding your suggestion. Please see line 426.

Lines 497-500 add a reference to this statement

Comment: We add this information to the manuscript. Please see line 545.

All authors contributed to general concept and design and writing of the article, and contributed to read, and approved the submitted version.

We are looking forward to your comments and critiques.

Sincerely,

Ines Krohn

July 7, 2022

Dr. Ines KROHN
University of Hamburg, Institute of Plant Science and Microbiology
Microbiology and Biotechnology
Ohnhorststr. 18
Hamburg 22609
Germany

Re: Spectrum00633-22R1 (Microalgae and bacteria interaction - Evidence for division of diligence in the alga microbiota)

Dear Dr. Ines KROHN:

Your manuscript has been accepted, and I am forwarding it to the ASM Journals Department for publication. You will be notified when your proofs are ready to be viewed.

Sincerely,

Eva Sonnenschein
Editor, Microbiology Spectrum
